# Reduced-Pressure Process for Fabricating Tea Tree Oil—Polyvinylpyrrolidone Electrospun Fibers

**DOI:** 10.3390/polym14040743

**Published:** 2022-02-15

**Authors:** Li Zhu, Siti Machmudah, Hideki Kanda, Motonobu Goto

**Affiliations:** 1Department of Materials Process Engineering, Nagoya University, Furo-cho, Chikusa-ku, Nagoya 464-8603, Japan; zhu.li@g.mbox.nagoya-u.ac.jp (L.Z.); wahyudiono@b.mbox.nagoya-u.ac.jp (W.); 2Department of Chemical Engineering, Institut Teknologi Sepuluh Nopember, Surabaya 60111, Indonesia; machmudah@chem-eng.its.ac.id

**Keywords:** electrospinning, reduced pressure, evaporation, tea tree oil

## Abstract

Electrospun fibers containing tea tree oil (TTO) can be explored for practical applications due to the antimicrobial and anti-inflammatory activities of TTO. Considering that there are potentially toxic components in TTO, it is necessary to eliminate or reduce its content in the preparation process of TTO-doped electrospun fibers. In this work, electrospun TTO-PVP (polyvinylpyrrolidone) fibers containing an 18.18 wt.% decreased content of 1,8-Cineole were successfully fabricated by intense evaporation of a self-made reduced-pressure electrospinning (RP-ES) setup (as low as 94.4 kPa). In addition, such intense evaporation led to a morphology change, where a typical average fiber diameter increased from 0.831 to 1.148 μm, fewer and smaller beads in fibers, along with a rougher and grooves fiber surface. These morphology changes allowed Terpinen-4-ol to remain in the fiber for a more extended period. In addition, RP-ES proved the possibility for intense evaporation and continuous vapor removal by continuously environmental vacuum pumping of electrospinning.

## 1. Introduction

Tea tree oil (TTO) is a volatile essential oil extracted from Australian Melaleuca alternifolia by steam distillation and is valued globally for its antimicrobial and anti-inflammatory activities [1,2]. There have been works of literature focusing on the prominent effects of TTO and the synthesis of TTO-doped composites. For example, Ge et al. [3] reported dispersed and immersed TTO emulsion droplets in chitosan films for wound healing applications. Silva et al. [4] prepared tea tree oil-loaded core–shell nanocapsules for acne treatment. El-Wakil et al. [5] reported antimicrobial bio-composites based on rice bran with TTO with excellent possibility of food packaging and biomedical purposes.

Here, Terpinen-4-ol is the primary functional component of TTO and shows promising antibacterial [6,7,8], anti-inflammatory [9], anti-mite [10], and anti-lice [11] properties. However, 1,8-Cineole, another component of TTO, is responsible for potential skin irritation [12] and potential toxicity of kidneys [13] and fetal [14]. The TTO standard proposed by the International Organization for Standardization (ISO) proposed stipulates a limit for the 1,8-Cineole concentration in products, and selective removal of 1,8-Cineole from TTO is necessary for safe and comfortable use of TTO [15].

A possible solution to the selective removal of 1,8-cineole from TTO is based on the volatility differences. It has been reported that the boiling point of 1,8-Cineole (176–177 °C) is lower than Terpinen-4-ol (212 °C). In addition, 1,8-Cineole shall have a higher vapor pressure (1.90 mmHg at 25 °C) than Terpinen-4-ol (0.048 mmHg at 25 °C) [16,17,18,19]. This assumption has been experimentally proved by literature work as well. It has been reported by Leach et al. [20], that 1,8-Cineole has higher volatility, and with compared, Terpinen-4-ol has relatively low volatility. In this case, it is possible to control and adjust the content of 1,8-Cineole and Terpinen-4-ol in TTO-doped materials if the environmental pressure during material synthesis shall be reduced.

This work was to fabricate microscale TTO-doped fibers using electrospinning technology, which is widely studied and future industrialized with various applications such as drug delivery, medicine, sensors, and cosmetics. Here, fast evaporation is essential for the electrospinning process [21,22,23]. This paper proposed a self-design reduced-pressure electrospinning (RP-ES) setup was to enhance the evaporation and continuous vapor removal ability of the polymer-solvent and volatile components. The reduced pressure is based on a vacuum pump that continuously vacuums the chamber with a considerable size (0.15 m^3^) where RP-ES is located. Through our tests, this setup successfully fabricated different electrospun fiber mats. Upon a reduced pressure, more 1,8-Cineole volatilized of jetting micron-sized droplets, resulting in a high content of the Terpinene-4-ol solution, which could improve the application quality of TTO-doped electrospun fibers.

For practical TTO-doped fiber composites, there are different polymer hosts such as TTO-polylactic acid (PLA) [24], TTO-polyethylene oxide (PEO) [25,26], and TTO-polyurethane (PU) [27]. This study selected polyvinylpyrrolidone (PVP) as the polymer host to fabricate TTO-PVP electrospun fibers. PVP has a wide range of safe applications among various polymers in medicine, pharmacy, cosmetics, and industrial fabrication because it is non-toxic, physiologically compatible, temperature resistant, pH stable, chemically inert [28].

## 2. Materials and Methods

### 2.1. Materials

Tea tree oil (TTO) was obtained from Guangdong Academy of Forestry (Guangzhou, China), and GC-MS analyzed it according to Tranchida et al. [19]. The TTO comprised 38.68% Terpinen-4-ol and 4.29% 1,8-Cineole, which was complied with ISO 4730:2017. Polyvinylpyrrolidone (PVP average MW 1,300,000) was purchased from Sigma-Aldrich (St. Louis, MO, USA). Ethanol (99.5%) was purchased from Wako Pure Chemical Industries (Osaka, Japan). 

For the quantitative analysis of TTO components, the 1,8-Cineole standard (99%, CAS: 470-82-6, Catalog Number: C80601) was purchased from Sigma-Aldrich (St. Louis, MO, USA). The Terpinen-4-ol standard (CAS: 562-74-3, Catalog Number: T117500) was purchased from Toronto Research Chemicals (Toronto, ON, Canada).

### 2.2. Methods

#### 2.2.1. Experimental Setup

As described in Figure 1, a typical electrospinning system was placed in a 0.15 m^3^ acrylic chamber connected to a rotary vane vacuum pump (GHD-030, ULVAC, Inc., Miyazaki, Japan ) and a ball valve controlling the pressure from atmospheric (ATM) pressure (101.3 kPa) to reduced-pressure (RP) 94.4 kPa. The flow rate was set as 0.4 L/min, and fresh air from the airflow meter was set to remove solvent vapor. A high-pressure syringe pump (PHD-Ultra 4400, Harvard Apparatus, Holliston, USA) was injected into a 0.25 mm needle at a 0.1 mL/min flow rate. A DC voltage of 12 kV power supply (HARb-30P1, Matsusada Precision, Osaka, Japan) applied an electric field to the TTO-PVP polymer solution to produce a jet that accelerated from the needle to the collector at a distance of 12 cm [29]. The above experiments were carried out at 22–24 °C and 21–28% relative humidity using a digital thermo-hydro indicator (THI-HP, AS ONE Corporation, Osaka, Japan).

#### 2.2.2. GC-MS Analysis

The phytochemical parameters of the TTO, pure PVP and TTO-PVP electrospun fibers were identified by Gas Chromatography-Mass Spectrometry (GC-MS; Agilent 7890A GC and Agilent 5975C MS; Agilent Technologies, Inc., Santa Clara, CA, USA) and an HP-5ms capillary column (30 mm, 0.25 mm i.d., 0.25 μm film thickness; Agilent Technologies, Inc., Santa Clara, CA, USA). The chromatograph was programmed from 50 to 280 °C at a rate of 3 °C/min. Helium was used as the carrier at a 24 mL/min flow rate.

As shown in Figure 2, the absolute calibration curves between peak area and concentration of 1,8-Cineole and Terpinen-4-ol were plotted, and the four-point quantitative linear correlation coefficients (R2) were 0.9959 and 0.9975, respectively.

#### 2.2.3. Antibacterial Test

The density of *Staphylococcus aureus* (NBRC^®^ 12732™) strains suspension was grown in the Trypticase Soy Broth and adjusted by using a UV-vis spectrophotometer (V-550, JASCO, Tokyo, Japan) for 0.5 McFarland Standard with 0.1 reading absorbance at 600-nm [30]. The 10-day room condition exposed electrospun nanofiber discs (20 mg) were placed on Muller Hinton Agar and incubated at 37 °C medium for 48 h.

#### 2.2.4. Other Characterizations

The TTO, pure PVP, and TTO-PVP electrospun fibers were characterized using scanning electron microscope (SEM; S-4300, Hitachi, Tokyo, Japan) with a gold sputtering coating (RMC-Eiko RE vacuum coater, Eiki Engineering Co., Ltd., Ibaraki, Japan), thermogravimetric–differential thermal analysis (TG-DTA; TGA-50, Shimadzu, Kyoto, Japan), differential scanning calorimetry (DSC; DSC-60A, Shimadzu, Kyoto, Japan), and Fourier transform infrared spectroscopy at 4 cm^−1^ resolution (FT-IR; PerkinElmer Ltd., Waltham, MA, USA).

## 3. Results and Discussion

### 3.1. Fibers Morphologies

#### 3.1.1. The Ratio of TTO to PVP

PVP was first dissolved in ethanol for the polymer solution at a concentration of 10 wt.% based on previous research [31,32]. As shown in Figure 3, TTO-PVP electrospun fiber mats were prepared under an atmospheric environment. When the TTO concentration (of polymer solution) was higher than 15 wt.%, the electrospun fiber mat was deformed and became a transparent slimy film. The TTO of 10 wt.% or less could obtain electrospun fiber. Therefore, 10% by weight has a relatively high TTO percentage for the functional TTO-PVP electrospun fiber mats. The weight percentage of TTO:PVP = 1:1 electrospun fibers was to study for the following research.

#### 3.1.2. Fibers Formation by RP-ES

Figure 4 and Figure 5 showed that pure PVP and TTO-PVP electrospun fiber mats could be fabricated with 7–10 wt.% pure PVP and 7–10 wt.% TTO-PVP ethanol-based polymer solution at the environmental pressures from atmospheric 101.3 kPa to reduced-pressure 94.4 kPa. It is viewed from SEM images that the individual electrospun fibers were randomly distributed on the supporting base, with no branch structure or large knots. It also supported the feasibility of operating reduced pressure electrospinning to obtain the micro-scale fibers. 

#### 3.1.3. Fibers Diameter by RP-ES

The averaged fiber diameter was measured by SEM image using ImageJ 1.51 w at 150 randomly selected fibers. As shown in Figure 6, the average diameters of 7, 8, 9, 10 wt.% TTO and PVP electrospun fibers were increased as the pressure decreased with intense vacuum pumping. For example, the diameter of 10 wt.% TTO-PVP electrospun fiber increased from 0.831 μm of atmospheric 101.3 kPa to 1.148 μm of reduced pressure 94.4 kPa. This result could also indicate that if an electrospun fiber mat product with a smaller diameter is required, the wt.% of PVP in the polymer solution in RP-ES could be appropriately reduced.

#### 3.1.4. Less Bead-Fibers by RP-ES

For 8 wt.% pure PVP and 8 wt.% TTO-PVP electrospun fibers, it has been observed that the typical atmospheric electrospun of Figure 7a,e usually have the largest size and more micron-sized beads. The size and number of beads decrease with the enhanced vacuum pumping of Figure 7b–d,f–h. Bead-fibers are related to the concentration of PVP. When the PVP concentration is low, micron beads appear because of their low viscosity and cannot overcome the surface tension of the solution [33]. Increasing the concentration of PVP can avoid the appearance of bead-fibers. It also shows that RP-ES increases the viscosity of the polymer solution through the evaporation of the solvent.

#### 3.1.5. Non-Smooth Fibers Surface by RP-ES

10 wt.% TTO-PVP electrospun fibers were obtained at atmospheric of Figure 8a and reduced-pressure of Figure 8b–d conditions. It shows that mostly smooth surface fibers were obtained during the electrospinning process conducted at atmospheric pressure. Conversely, this fiber surface morphology became unsmooth or corrugated when the process was operated at low operating pressures of RP-ES. Due to the rapid evaporation, low operating pressure seems to cause wrinkles on the fiber surface of Figure 8b–d. When this phase separation occurred quickly, the solidified fiber skin was formed during fiber elongation before the polymer-solvent and volatile components of TTO were removed. Then, it diffused out of the fiber skin slowly, resulting in the non-smooth rougher and grooves surface morphology of the electrospun fiber [34,35,36,37].

### 3.2. Fibers Components Analysis

#### 3.2.1. TTO-PVP Fibers Compounds

The TTO essential oil extracted by steam distillation is the most widely used aromatic compound, containing various single molecular chains, molecular olefins and other chains, aromatic compounds, and alcohol groups. Here, each 2 mg of 10 wt.% TTO-PVP electrospun fiber mats were dissolved in 1.5 mL ethanol and injected via an autosampler device into the GC-MS apparatus to observe the TTO component contents. The original TTO was dissolved in ethanol and injected into the GC-MS device as a reference. Table 1 shows the list of main TTO components that remained in the TTO-PVP electrospun fiber products fabricated under atmospheric 101.3 kPa and reduced-pressure 94.4 kPa conditions. The result indicates that volatile components (the components before Terpinolene of Peak No. 7 measured by GC-MS) more evaporated during micron-sized polymer solution droplets jettied at high speed, remaining a high concentration Terpinen-4-ol of low volatility.

As shown in Figure 9a,b of peak areas, the content of 1,8-Cineole (Peak 1) in TTO-PVP electrospun fiber mats fabricated by RP-ES decreased by 18.18 wt.% (Peak area decreased from 58,909,143 to 48,920,123 units), while the main functional component, Terpinen-4-ol (Peak 2), only reduced by 2.05 wt.% (Peak area decreased from 744,520,273 to 720,972,528 units). This TTO-PVP electrospun fiber with a lower 1,8-Cineole concentration could have a safer application prospect. It also shows that RP-ES has the intense ability to evaporate the volatiles of the mixed solution.

#### 3.2.2. Fibers Exposure Test

In the electrospun fiber mats placement test, each TTO-PVP electrospun fiber (12–14 mg) was placed to ambient settings of 14–16 °C and 35–40% humidity to test the encapsulation ability of Terpinen-4-ol and 1,8-Cineole with PVP. It can be observed from Figure 10 and calculated by the standard Linear Calibration Curve of Figure 2 that at the beginning (0 days), Terpinen-4-ol of the TTO-PVP fiber electrospun at atmospheric pressure (0.391 (%*w*/*w*)) was slightly higher than that of the RP electrospun fiber (0.383 (%*w*/*w*)). After 10 d of aging, there was still a Terpinen-4-ol residual of 0.058 mg/mg in the fiber electrospun at atmospheric pressure, while the Terpinen-4-ol residual in the RP-ES fiber was 0.093 mg/mg (60.34 wt.% higher). Because the fiber electrospun with RP-ES was thicker than that at atmospheric pressure, the Terpinen-4-ol inside the fiber was more difficult to evaporate in an exposed environment. In contrast, when using typical electrospinning or RP-ES, 1,8-Cineole vanished in the exposed TTO-PVP electrospun fiber on day 5 and later analysis. It indicated that 1,8-Cineole could not combine well with PVP. In addition, under room conditions, 1,8-Cineole was far more volatile than Terpinen-4-ol, which supports the hypothesis that RP-ES could evaporate and remove more 1,8-Cineole than typical electrospinning methods. It is viewed that RP-ES can produce a lower-content 1,8-Cineole of potentially toxic in TTO-PVP electrospun fibers, thus improving the application safety. In addition, GC-MS results showed that 10 wt.% of TTO-PVP electrospun fibers fabricated by RP-ES contained Terpinen-4-ol after placing for 10 days.

*Staphylococcus aureus* (*S. aureus*) is one of the most common bacterial infections in humans and is the causative agent of many human infections [38]. Terpinen-4-ol exhibits a strong ability of antibacterial and antibiofilm against *S. aureus* [39]. The experimental results showed that the 10 wt.% TTO-PVP electrospun fibers after-10-day placement (Figure 11b) had approximately 3 times the inhibition zone area on the agar compared to the pure PVP electrospun fibers (Figure 11a). The results showed that the 10 wt.% TTO-PVP electrospun fibers fabricated by RP-ES had the ability to resist *S. aureus* for 10 days.

### 3.3. Fibers Properties Analysis

#### 3.3.1. FT-IR Analysis

Figure 12 shows that FTIR spectroscopy could detect chemical bonds between unknown materials and compounds in these contents. Therefore, the chemical structure changes of PVP molecules with or without TTO could be observed after electrospinning. As illustrated in curve (a), for the original TTO, a strong peak at 2962 cm^−1^ was detected. It was ascribed to the stretching vibration peak of the C–H bond. Absorption bands at 1126 cm^−1^, related to the stretching vibration of the C–O bond of the tertiary alcohol in terpenes and terpineol, were observed [40]. This FTIR spectrum also shows a peak at 3450 cm^−1^, ascribed to the O–H bond stretching vibration, and the 887 cm^−1^, 864 cm^−1^, and 799 cm^−1^ regions are ascribed to the Terpinen-4-ol compound [41]. In curve (b) for PVP fibers, the IR peaks located at 3410 cm^−1^, 2954 cm^−1^, 1654 cm^−1^, 1422 cm^−1^, 1288 cm^−1^, and 841 cm^−1^ were assigned to the stretching vibrations of the O–H, C–H, C=O, C=C, C–N, and =C–H groups, respectively [42,43]. The absorption peaks of Terpinen-4-ol were found at 887 cm^−1^, 864 cm^−1^, and 799 cm^−1^ [41]. These results indicate that TTO has embedded PVP, and the electrospinning process at reduced pressure did not shift the properties of PVP and TTO as starting materials.

#### 3.3.2. Thermal Properties

Figure 13 shows the curves of thermogravimetric–differential thermal analysis (TG-DTA) under a nitrogen flow rate of 50 mL/min with 5 °C/min increased from 40 to 500 °C, where the weight shift during the analysis process was used to understand the thermal behavior of the materials and their volatile component fractions. As shown in curve (a), PVP electrospun mats show weight loss at 40–100 °C, attributed to moisture evaporation. The primary weight loss of 380–460 °C could be attributed to the standard thermal decomposition of PVP. The pure TTO of the curve (e) was evaporated completely before 125 °C. As shown in curves (b) and (c), electrospun fibers containing 10 wt.% TTO have three-step weight loss, namely below 175 °C, 175–370 °C, and over 370 °C. There was no apparent difference between the two curves. When the amount of TTO in the PVP solution as the starting material increases to 15% of the curve (d), the curve changes more obviously. Its results show that consistent with infrared spectroscopy analysis (see Figure 12), TTO is present in PVP electrospun fiber products. It may change the thermal stability of electrospun products through the interaction of C-H and C=O bond conjugation [44]. It finally indicated RP-ES could successfully fabricate the 10 wt.% TTO-PVP of polymer solution electrospun fiber mats.

Figure 14 shows the curves of differential scanning calorimetry (DSC) under a Nitrogen flow rate of 50 mL/min with 10 °C/min increased from 0 to 100 °C. It can be seen that the endothermic peak of the TTO-PVP electrospun fibers (curves (a, b)) is about 15 °C lower than that of the PVP electrospun fibers (curves (c, d)). This may be because the TTO in the TTO-PVP electrospun fibers volatilized when heated to ~60 °C resulting in the need for endothermic heat of the fibers mat. In addition, probably because more volatilized components such as 1,8-Cineole were already evaporated during the preparation of fibers by RP-ES, the endothermic peak of TTO-PVP by RP-ES (curve (a)) was 1 °C higher than by typical electrospinning (curve (b)). Similarly, for the pure PVP electrospun fibers produced by RP-ES (curve (c)), the evaporation of the solvent could be more intense, resulting in drier fiber mats, causing the endothermic peak to shift 1.5 °C higher than by typical electrospinning. It further supported the enhanced evaporation of RP-ES for polymer solvents and volatile components.

### 3.4. Mechanism of Enhanced Evaporation by RP-ES

The above results indicate that the evaporation of polymer-solvent and TTO volatile components in RP-ES was enhanced. This paper attempts to illustrate the mechanism of enhanced evaporation of polymer-solvent ethanol. As shown in Figure 15a,b the average angle of the Taylor cone increased from 72° to 76°, which may be attributed to the stronger evaporation of ethanol in the Taylor cone area that increased the concentration PVP in the polymer solution at the Taylor cone region. As the viscosity of polymer solution increases, resulting in stronger cohesion between the PVP chains and making it impossible to stretch the molecules further that increased fiber diameter [45,46,47].

As shown in Figure 15a,c during the electrospinning jet, as the total pressure of the electrospinning ambient environment decreases by continuous vacuum pumping, the actual vapor pressure of the polymer-solution liquid surface is simultaneously reduced by the partial pressures of Dalton’s law. According to Dalton’s law, the total pressure *P_tot_* is detailed as,
(1)$$P_{tot\;}=P_{EtOH}+P_{air}$$
(2)$$P_{tot\;}=nP_{EtOH}$$

It leads to an enlarged polymer-solvent pressure difference Δ*P_EtOH_* between the saturated vapor pressure *Ps_EtOH_* and actual vapor pressure *P_EtOH_* at a fixed temperature, which results in intense vapor diffusion. The pressure difference Δ*P_EtOH_* is detailed as,
(3)$$\Delta P_{EtOH}=Ps_{EtOH\;}-P_{EtOH\;}$$
with the greater Δ*P_EtOH_*, the evaporation shall be faster. Simultaneously, from the relative humidity (*RH*) perspective, the partial water vapor pressure *P* decreases because water molecules are continuously vacuum pumping, causing an increased water pressure difference Δ*P* as,
(4)ΔP=Ps−P=Ps (1−RH)

Here, *Ps* is the saturated water vapor pressure. When the water vapor molecules were pumped away, the dropped in *P* led to the dropped in *RH*. Several recent studies have identified ambient *RH* is critical in controlling aqueous polymer-solvent evaporation during electrospinning [48,49,50]. Pelipenko et al. [51] and Vrieze et al. [52], found that a decreasing *RH* led to an increased diameter of PVP electrospun fibers. Here, since PVP can be dissolved in water, the moisture absorption around PVP was less under a low *RH*. Therefore, as the PVP solidified, the electrospun fiber cannot be elongated anymore, resulting in thicker fibers [53], which agrees with our observations in this study.

In addition, Figure 15c shows that the fresh airflow by continuous vacuum pumping could blow away the polymer-solvent vapor and maintain a significant vapor concentration difference that tends to benefit a long-time working electrospinning system.

## 4. Conclusions

Compared with typical electrospinning methods, RP-ES can enhance the evaporation of the polymer-solvent, thereby increasing the viscosity of the polymer solution in the Taylor cone region and the process of jetting. This results in (1) thicker fibers, (2) fewer and size reduced bead of fibers, and (3) non-smooth grooves fiber surface morphology. FT-IR and TG results showed that RP-ES did not change functional groups and thermal properties of TTO-PVP electrospun fibers. In addition, with an 18.18 wt.% decreased concentration of 1,8-Cineole, making this product safer for application. Based on the results and owing to the enhanced evaporation capacity of RP-ES, it seems that, by reduced pressure and continuous vacuum pumping to remove water and solvent vapor, RP-ES tends to keep working longer than current electrospinning methods.

## Figures and Tables

**Figure 1 polymers-14-00743-f001:**
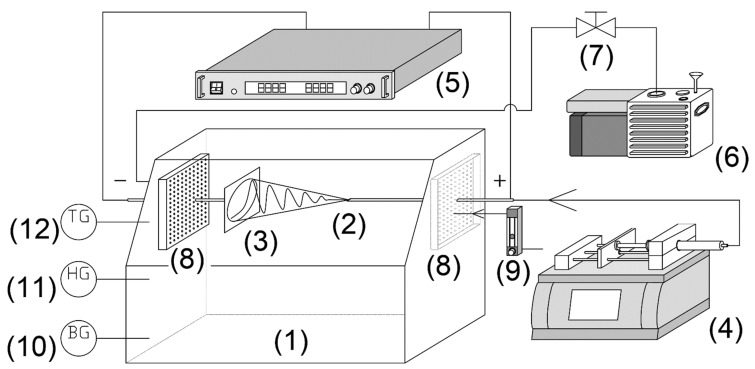
Experimental setup for electrospinning; (**1**) 0.15 m^3^ acrylic chamber, (**2**) nozzle, (**3**) fiber collector, (**4**) high-pressure syringe pump, (**5**) high-voltage power supply, (**6**) vacuum pump, (**7**) pressure control ball-valve, (**8**) baffle plate, (**9**) airflow meter, (**10**) barometer gauge, (**11**) humidity gauge, (**12**) temperature gauge.

**Figure 2 polymers-14-00743-f002:**
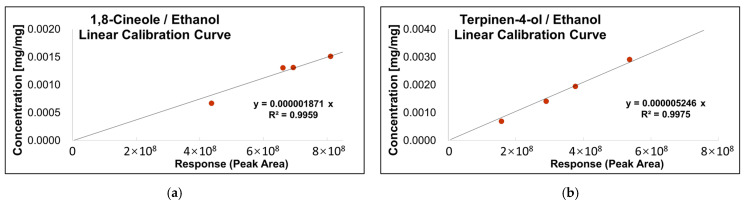
Linear calibration curve of the relationship between peak area and concentration with (**a**) 1,8-Cineole, and (**b**) Terpinen-4-ol by GC-MS.

**Figure 3 polymers-14-00743-f003:**
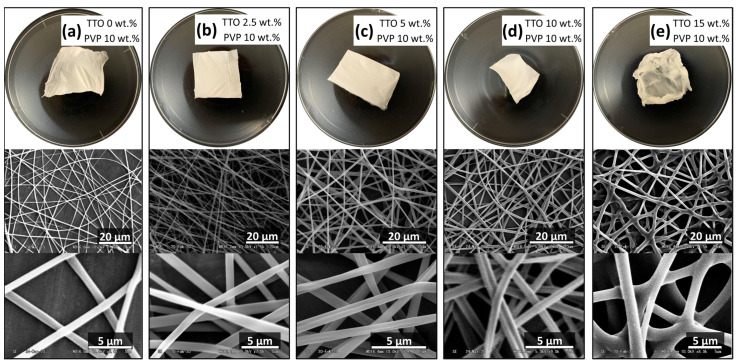
Appearances and SEM images of 10 wt.% PVP fiber mat by typical atmospheric electrospinning with (**a**) 0 wt.%, (**b**) 2.5 wt.%, (**c**) 5 wt.%, (**d**) 10 wt.%, and (**e**) 15 wt.% TTO.

**Figure 4 polymers-14-00743-f004:**
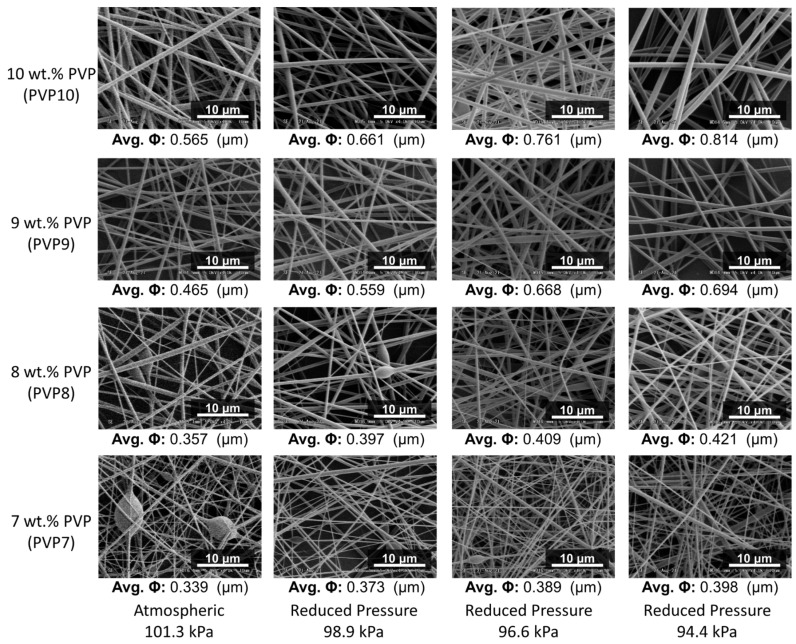
SEM images and average fiber diameter of pure PVP electrospun fibers.

**Figure 5 polymers-14-00743-f005:**
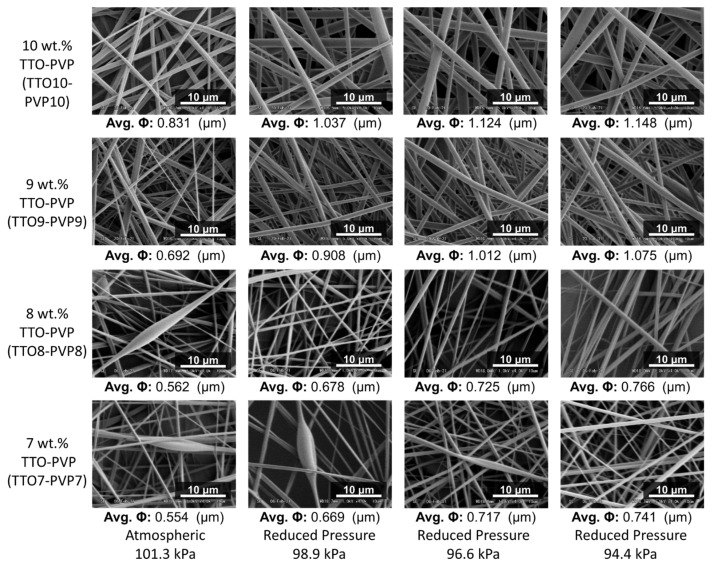
SEM images and average fiber diameter of TTO-PVP electrospun fibers.

**Figure 6 polymers-14-00743-f006:**
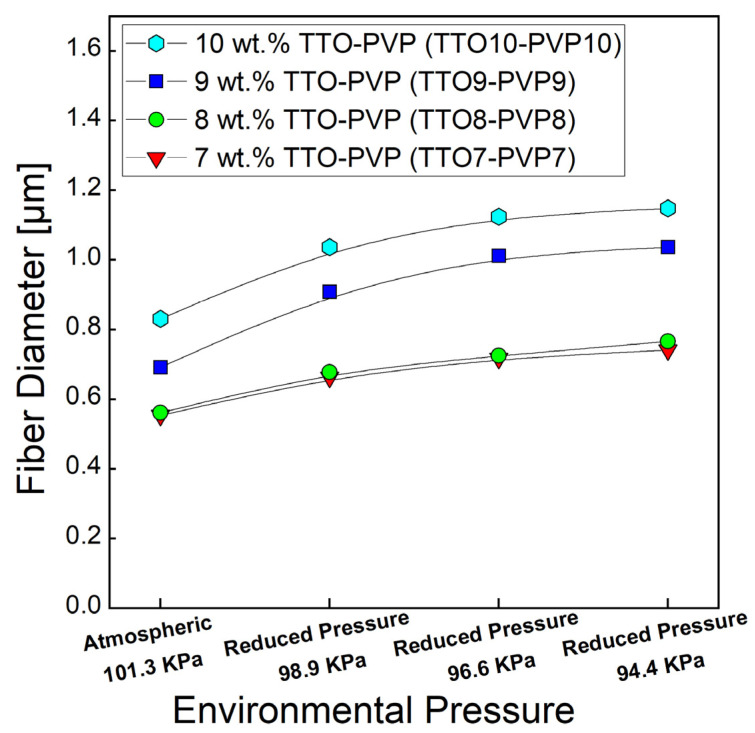
Distribution of TTO-PVP electrospun fibers diameter.

**Figure 7 polymers-14-00743-f007:**
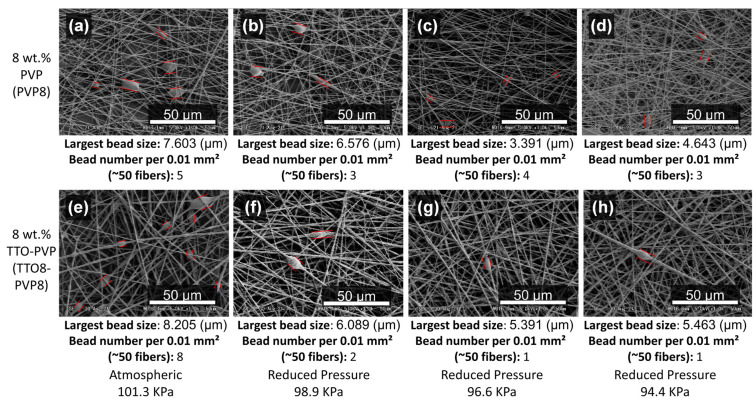
SEM images of PVP8 electrospun fibers fabricated at (**a**) atmospheric 101.3 kPa, reduced pressure of (**b**) 98.9 kPa, (**c**) 96.6 kPa, (**d**) 94.4 kPa, and TTO8-PVP8 electrospun fibers fabricated at (**e**) atmospheric 101.3 kPa, reduced pressure of (**f**) 98.9 kPa, (**g**) 96.6 kPa, (**h**) 94.4 kPa.

**Figure 8 polymers-14-00743-f008:**
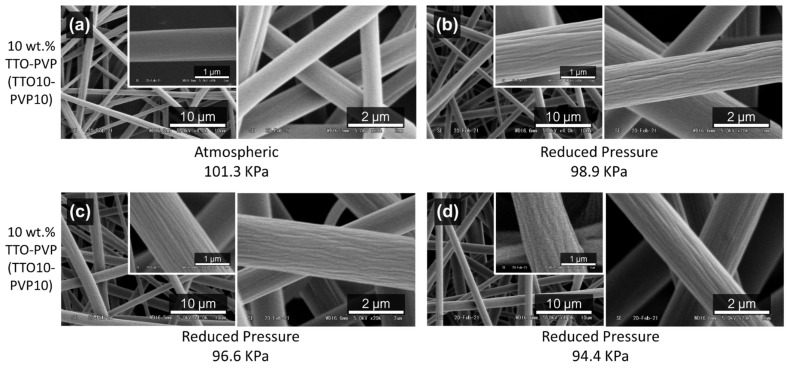
SEM images of TTO10-PVP10 electrospun fibers surface fabricated at (**a**) atmospheric 101.3 kPa, reduced pressure of (**b**) 98.9 kPa, (**c**) 96.6 kPa, (**d**) 94.4 kPa.

**Figure 9 polymers-14-00743-f009:**
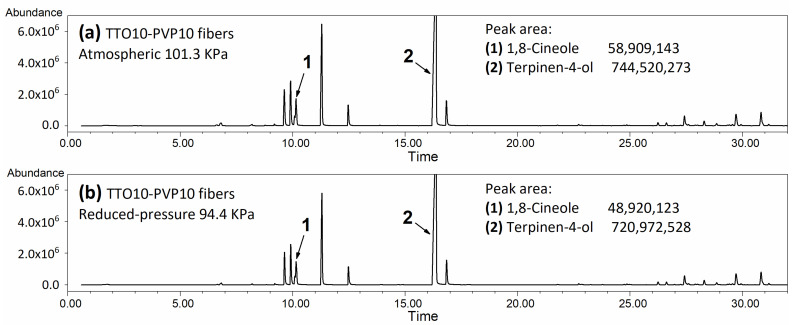
GC-MS traces of TTO chemotypes in 10 wt.% TTO-PVP electrospun fibers fabricated at (**a**) atmospheric 101 kPa, (**b**) reduced pressure 94.4 kPa.

**Figure 10 polymers-14-00743-f010:**
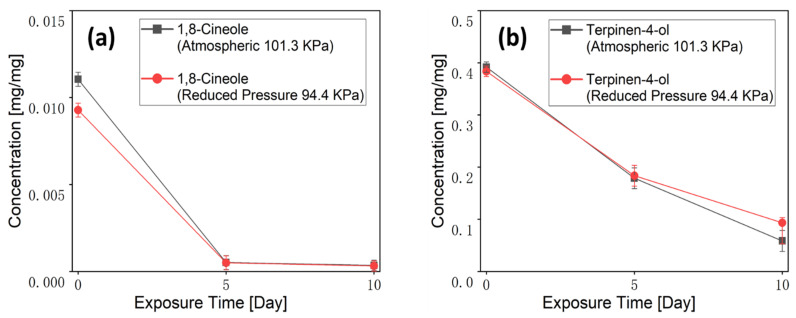
Concentrations of (**a**) 1,8-Cineole, and (**b**) Terpinen-4-ol in the TTO-PVP of exposed electrospun fiber. The concentrations were obtained by GC-MS analysis.

**Figure 11 polymers-14-00743-f011:**
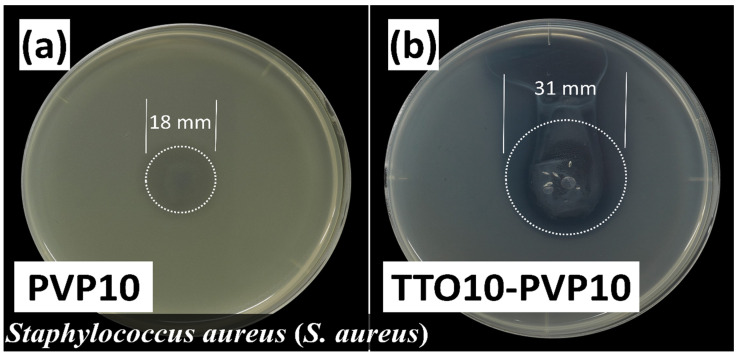
Inhibition zone of *Staphylococcus aureus* using after-10-day room condition placement of (**a**) 10 wt.% PVP (**b**) 10 wt.% TTO-PVP electrospun fibers fabricated by RP-ES.

**Figure 12 polymers-14-00743-f012:**
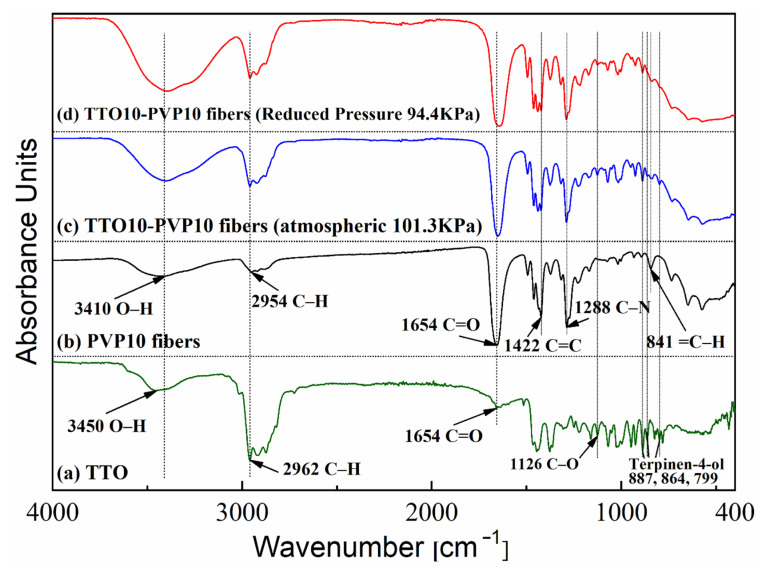
FTIR spectra of TTO10-PVP10 electrospun fibers fabricated at (**a**) reduced pressure 94.4 kPa, (**b**) atmospheric 101.3 kPa, (**c**) PVP10 electrospun fibers of atmospheric 101.3 kPa, and (**d**) TTO.

**Figure 13 polymers-14-00743-f013:**
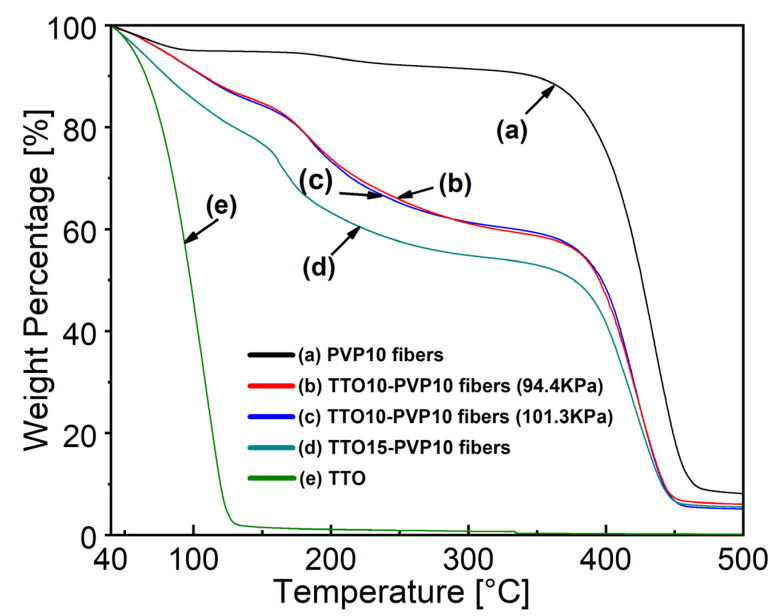
TGA thermogram of (a) PVP10 electrospun fibers of atmospheric 101.3 kPa, TTO10-PVP10 electrospun fibers fabricated at (b) reduced pressure 94.4 kPa, (c) atmospheric 101.3 kPa, (d) TTO15-PVP10 electrospun fiber of atmospheric 101.3 kPa, and (e) TTO.

**Figure 14 polymers-14-00743-f014:**
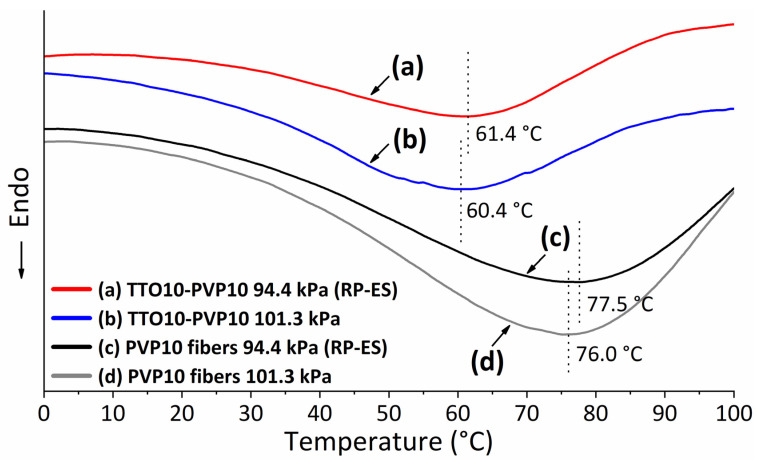
DSC analysis of TTO10-PVP10 electrospun fibers fabricated at (a) reduced pressure 94.4 kPa, (b) atmospheric 101.3 kPa, and PVP10 electrospun fibers fabricated at (c) reduced pressure 94.4 kPa, (d) atmospheric 101.3 kPa.

**Figure 15 polymers-14-00743-f015:**
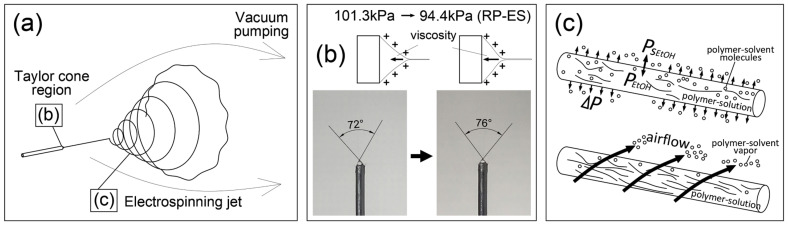
(**a**) Reduce pressure electrospinning process; (**b**) evaporation of Taylor cone area; (**c**) evaporation of electrospinning jet.

**Table 1 polymers-14-00743-t001:** Main Volatile Chemotypes of TTO Determined by GC-MS.

Peak No.	Compound Name	Retention Time [min]	Boiling Point [°C]	Peak Area in Concentration [%] *		
				ATM	RP	TTO
1	α-Pinene	6.84	155–156	0.72	0.45	2.28
2	α-Terpinene	9.64	173–174	5.37	4.57	10.63
3	p-Cymene	9.95	177	6.21	5.58	4.23
4	β-Phellandrene	10.11	171–172	1.04	0.87	1.53
5	1,8-Cineole	10.16	176–177	3.92	3.59	4.29
6	γ-Terpinene	11.29	183	15.42	14.04	21.59
7	Terpinolene	12.52	184–185	2.90	2.69	3.60
8	Terpinen-4-ol	16.30	211–213	49.60	52.86	38.68
9	α-Terpineol	16.85	214–217	3.62	4.11	2.82
10	Aromadendrene	27.44	258–259	1.84	1.87	1.23
11	Alloaromadendrene	28.32	265–267	0.82	0.85	0.55
12	Ledene	29.73	268–270	2.28	2.39	1.61
13	δ-Cadinene	30.85	279–280	2.66	2.78	1.81
	others			3.61	3.33	5.15

* Concentrations as relative % of peak-area calculation from GC-MS; ATM: TTO10-PVP10 electrospun fiber fabricated at atmospheric pressure (101.3 kPa); RP: TTO10-PVP10 electrospun fiber fabricated at reduced-pressure (94.4 kPa); TTO: Tea Tree Oil.

## Data Availability

Not applicable.

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
