# Peer review of "Reduced-Pressure Process for Fabricating Tea Tree Oil—Polyvinylpyrrolidone Electrospun Fibers"

_polymers, 2022, doi:10.3390/polym14040743_

Round 1

Reviewer 1 Report

Although the paper is completely interesting, authors should improve it according to following lines:

1- Authors should provide mechanical analysis to understand the effect of their process on the mechanical characteristics of resultant fibers. This test also has relationship with physical properties. Results should also compared with those with no evaporation.

2- Authors should conduct DSC test to understand the thermal characteristics of fibers with and without evaporation.

3- As previously proved by researches, TTO has several medical applications. However, authors did not conduct any test to compare (with and without evaporation) the effect of their process on different biomedical properties such as antimicrobial and cytotoxicity.

Author Response

Sincerely thank you very much for your questions and suggestions for improving the manuscript.

1- Authors should provide mechanical analysis to understand the effect of their process on the mechanical characteristics of resultant fibers. This test also has relationship with physical properties. Results should also compared with those with no evaporation.

We thank the reviewer for the suggestion. Regarding this point, when we have used electrospinning in previous papers to create fibers with different functions to those in this study, we have not been asked to include mechanical properties. For example, our previous paper (https://doi.org/10.1016/j.reactfunctpolym.2019.05.016) was published in Reactive and Functional Polymers (impact factor=3.975) without mechanical.

On the other hand, in a previous paper by other researchers (https://doi.org/10.1016/j.aej.2016.04.025) (Alexandria Engineering Journal, impact factor=3.732), the mechanical properties have been examined, however the fibers examined in the study were made by typical, common electrospinning.

It means that research on the strength of fibers and research on adding functions to fibers could be considered separately in this journal here. This means that if we were to carry out a study on the strength of the fibers produced in this study, it would be worthy of submission as a separate new paper.

2- Authors should conduct DSC test to understand the thermal characteristics of fibers with and without evaporation.

Thank you very much. We added DSC analysis for the PVP and TTO-PVP electrospun fiber mats in section 3.3.2 Thermal Properties. The DSC analysis results further indicated the enhanced evaporation of RP-ES for polymer solvents and volatile components. We added the below sentences and Figure 14.

2.2.4 Other Characterizations
……………… Differential Scanning Calorimetry (DSC; DSC-60A, Shimadzu, Japan), ……………

3.3.2 Thermal Properties.
……………………………………
Figure 14 shows the curves of Differential Scanning Calorimetry (DSC) under a Nitrogen flow rate of 50 mL/min with 10 °C/min increased from 0 to 100 °C. It can be seen that the endothermic peak of the TTO-PVP electrospun fibers (curves (a, b)) is about 15 °C lower than that of the PVP electrospun fibers (curves (c, d)). This may be because the TTO in the TTO-PVP electrospun fibers volatilized when heated to ~60 °C resulting in the need for endothermic heat of the fibers mat. Also, probably because more volatilized components such as 1,8-Cineole were already evaporated during the preparation of fibers by RP-ES, the endothermic peak of TTO-PVP by RP-ES (curve (a)) was 1 °C higher than by typical electrospinning (curve (b)). Similarly, for the pure PVP electrospun fibers produced by RP-ES (curve (c)), the evaporation of the solvent could be more intense, resulting in drier fiber mats, causing the endothermic peak to shift 1.5 °C higher than by typical electro-spinning. It further indicated the enhanced evaporation of RP-ES for polymer solvents and volatile components.

3- As previously proved by researches, TTO has several medical applications. However, authors did not conduct any test to compare (with and without evaporation) the effect of their process on different biomedical properties such as antimicrobial and cytotoxicity.

Thank you very much for your suggestion. We added previous antibacterial test results of anti-Staphylococcus aureus for TTO-PVP electrospun fiber mats after 10 days of placement in room conditions of section 3.2.2 Fibers Exposure Test. The experimental results show that the 10 wt.% TTO-PVP electrospun fibers fabricated by RP-ES still have the antibacterial property of Staphylococcus aureus after being placed for 10 days. We added the below sentences and Figure 11.

2.2.3 Antibacterial Test
The density of Staphylococcus aureus (NBRC® 12732™) strains suspension was grown in the Trypticase Soy Broth and adjusted by using a UV-vis spectrophotometer (V-550, JASCO, Japan) for 0.5 McFarland Standard with 0.1 reading absorbance at 600-nm [31]. The 10-day room condition exposed electrospun nanofiber discs (20 mg) were placed on Muller Hinton Agar and incubated at 37 °C medium for 48 h.

3.2.2 Fibers Exposure Test
……………………………………
Staphylococcus aureus (S. aureus) is one of the most common bacterial infections in humans and is the causative agent of many human infections [39]. Terpinen-4-ol exhibits a strong ability of antibacterial and antibiofilm against S. aureus [40,41]. The experimental results showed that the 10 wt.% TTO-PVP electrospun fibers after-10-day placement (Figure 11(b)) had approximately 3 times the inhibition zone area on the agar compared to the pure PVP electrospun fibers (Figure 11(a)). The results showed that the 10 wt% TTO-PVP electrospun fibers fabricated by RP-ES had the ability to resist S. aureus for 10 days.

Thanks again for your questions and suggestions.

Reviewer 2 Report

This manuscript prepared electrospun TTO-PVP (Tea Tree Oil-polyvinylpyrrolidone) microscale fibers containing an 18.18 wt.% decreased content of 1,8-Cineoleby by self-made reduced-pressure electrospinning (RP-ES) setup. The work is interesting and the main conclusions are supported by experimental results. However, there are some problems and detailed comments as follows:

  1. In Figure 3, why the lower right corner scale of different groups is inconsistent?

  1. In the article, it is mentioned that “Also, Terpinen-4-ol could firmly adhere to PVP fibers fabricated by RP-ES for more than 10 days.” However, in Figure 10, the concentration of Terpinen-4-ol decreased more than a half after 5 days.

  1. In this work, the boiling point of 1,8-Cineole (176-177 ℃) is lower than Terpinen-4-ol (212 ℃), namely 1,8-Cineole has higher volatility compared with Terpinen-4-ol. Thus the RP-ES can enhance the evaporation of the polymer-solvent, thereby can decreased the concentration of 1,8-Cineole. However, why not decrease the concentration of toxic 1,8-Cineole directly by distillation at first, then adopt electrospinning technology?

Author Response

Thank you very much for your questions and suggestions for improving the manuscript.

1. In Figure 3, why the lower right corner scale of different groups is inconsistent?

Thank you very much for your careful checking. We have replaced the image and added the appropriate scale bar in Figure 3. Thanks for checking again.

2. In the article, it is mentioned that “Also, Terpinen-4-ol could firmly adhere to PVP fibers fabricated by RP-ES for more than 10 days.” However, in Figure 10, the concentration of Terpinen-4-ol decreased more than a half after 5 days.

I am very sorry for this problem. We correct it as follows: 

3.2.2 Fibers Exposure Test
……………………………………
“Also, GC-MS results showed that 10 wt% of TTO-PVP electrospun fibers fabricated by RP-ES contained Terpinen-4-ol after placing for 10 days. ”

3. In this work, the boiling point of 1,8-Cineole (176-177 ℃) is lower than Terpinen-4-ol (212 ℃), namely 1,8-Cineole has higher volatility compared with Terpinen-4-ol. Thus the RP-ES can enhance the evaporation of the polymer-solvent, thereby can decreased the concentration of 1,8-Cineole. However, why not decrease the concentration of toxic 1,8-Cineole directly by distillation at first, then adopt electrospinning technology?

Thank you very much for your question. The content of 1,8-Cineole in Tea Tree Oil can be reduced using distillation. It is reported that by using molecular distillation, under the condition of a vacuum of 110 Pa, distillation temperature of 38 °C, it will be possible to distill part of 1,8-Cineole from 2.67 % to 1.41 % [Niu, B.; Liang, Y.; Liang, J.; Liu, Y. Study on the enrichment of characteristic components of tea tree oil by molecular distillation. Food Research and Development 2019, 40, 25-31]. 

However, in this report, the ~49.26 wt.% distilled light components contained ONLY 26.34 % Terpinen-4-ol. These products do not meet the requirement of ISO 4730 (≥35 % Terpinen-4-ol) and cannot be treated as Tea Tree Oil, resulting in waste. 

Also, since the toxicity of 1,8-Cineole is not significant, previous pieces of literature have shown that it was “potentially” toxic. Therefore, in the industrial production of Tea Tree Oil, there is no exceptional process to remove 1,8-Cineole. 

And RP-ES provides a convenient, one-step method to reduce 1,8-Cineole in TTO-PVP electrospun fibers, improving its application safety. In addition, this paper utilizes the reduced 1,8-Cineole content as an aid to demonstrate the enhanced evaporation of reduced pressure electrospinning.

Thanks again for your questions and suggestions.

Round 2

Reviewer 1 Report

Authors well responded my comments and it is acceptable now.

Reviewer 2 Report

Accepted at it is.